# Comprehensive Analysis of Antioxidant and Hepatoprotective Properties of *Morus nigra* L.

**DOI:** 10.3390/antiox12020382

**Published:** 2023-02-04

**Authors:** Saša Vukmirović, Vladimirka Ilić, Vanja Tadić, Ivan Čapo, Nebojša Pavlović, Ana Tomas, Milica Paut Kusturica, Nataša Tomić, Svetolik Maksimović, Nebojša Stilinović

**Affiliations:** 1Department of Pharmacology and Toxicology, Faculty of Medicine Novi Sad, University of Novi Sad, 21000 Novi Sad, Serbia; 2Department for Pharmaceutical Research and Development, Institute for Medicinal Plant Research “Dr. Josif Pancic”, Tadeusa Koscuska 1, 11000 Belgrade, Serbia; 3Department of Histology and Embryology, Medical Faculty of Novi Sad, University of Novi Sad, 21000 Novi Sad, Serbia; 4Department of Pharmacy, Faculty of Medicine Novi Sad, University of Novi Sad, 21000 Novi Sad, Serbia; 5Institute of Emergency Medicine, Clinical Center of Vojvodina, Novi Sad, 21000, Serbia; 6Faculty of Medicine Novi Sad, University of Novi Sad, 21000 Novi Sad, Serbia; 7Department of Organic Chemical Technology, Faculty of Technology and Metallurgy, University of Belgrade, Karnegijeva 4, P.O. Box 3503, 11120 Belgrade, Serbia

**Keywords:** oxidative stress, screening, chemical composition, nutraceuticals, hepatoprotective

## Abstract

The framework of this study was a comprehensive investigation of *Morus nigra* L. extracts, with the aim to establish the correlation between chemical composition and antioxidant/hepatoprotective activity of a series of black mulberry extracts obtained from aerial parts of the plant. Black mulberry leaf (MLEE), bark (MBEE), juice (MJ) and fresh fruit (MFEE) extracts were obtained using the conventional Soxhlet extraction, while the supercritical CO_2_ extraction procedure was employed for preparation of the seed oil (MSO). Analysis of the chemical composition was performed using spectrophotometric, HPLC and GC methods. For the evaluation of antioxidant activity, in vitro FRAP and DPPH assays were applied. In Haan strain NMRI mice with streptozotocin-induced oxidative stress, in vivo antioxidant activity and liver tissue integrity were examined. The content of polyphenolic compounds was the highest in MBEE (68.3 ± 0.7 mgGAE/g) with the most abundant compounds being polyphenolic acids, followed by MLEE (23.4 ± 0.5 mgGAE/g) with the flavonoids isoquercetin and rutin being present in a significant amount. An analysis of MSO revealed a high content of γ-linoleic acid. The highest antioxidant activity in vitro (FRAP and DPPH) was observed for MLEE, MBEE and MSO. Beneficial effects were confirmed in vivo, with lower values of hepatosomatic index, potentiation of the activity of the enzymes superoxide dismutase and catalase, a lower rate of lipid peroxidation and reduced positivity for the P450 enzyme in animals treated with MLEE, MBEE and MSO. Black mulberry leaf and bark extracts as well as seed oil exhibited significant antioxidant activity. Apart from the confirmed biological properties of the fruit and leaf extracts, the observed activities of black mulberry seed oil and bark extract imply its importance as a sustainable source of phytochemicals.

## 1. Introduction

Hepatotoxicity and liver disease are related to a variety of factors such as drugs, alcohol, obesity, viruses, autoimmune disorders, etc. [1]. Phase 1 of drug metabolism can result in the production of toxic metabolites, inducing the creation and accumulation of reactive oxygen species (ROS) [2,3]. In order to adapt to ROS exposure, living organisms produce antioxidant enzymes. However, when the free radicals and ROS levels surpass the antioxidant capability, oxidative stress occurs [4], leading to different ailments, including liver disease [5].

Conventional drug use in treating hepatic disorders can be hindered by challenges regarding their efficacy and safety. In these circumstances, herbal medicines, sometimes characterized by lower costs and a better safety profile in comparison to conventional drugs, offer a suitable alternative. Thus, the plant kingdom is considered an important source of compounds with therapeutic potential adopted for the alleviation of numerous diseases. This particularly refers to the comestible plants with proven antioxidant and hepatoprotective activities [1,6].

In recent years, berry fruits have gained the spotlight in nutritional research due to their numerous positive effects on human health [7]. *Morus nigra* L., usually called black mulberry, is a species belonging to the Moraceae family, which is native to southwestern Asia, but for centuries it has also been grown throughout Europe [8]. In Asia, mulberry was grown to support silk production, since mulberry leaves are considered as the major source of nutrients for silk worm (*Bombyx mori* L.) [9]. On the other hand, in most European countries, plants from this family were usually valued for their fruits, while leaves were considered waste or, in several countries including Serbia, have been traditionally used for tea preparation [10].

Recent research demonstrated that black mulberry has a wide range of pharmacological activities, such as antioxidant, antinociceptive, anti-inflammatory, antidiabetic, anticancer, antihyperlipidemic and antimicrobial activities as well as anti-obesity effects [11]. Potential medicinal use of black mulberry is mainly based on its rich content of phenols [12], compounds with vast structural diversity. Simple molecules such as phenolic acids (e.g., ferulic, gallic and caffeic) as well as polyphenolic compounds such as flavonoids and tannins exhibit numerous biological and pharmacological activities [5,13].

Although there is a wealth of literature about the beneficial effects of *M. nigra*, studies usually focus on a single part of the plant such as fruits [12,14,15,16,17,18], leaves [10,19,20,21] and bark [22,23]. In order to tackle this issue, we wanted to undertake a study to evaluate the effects of all aerial parts of the *M. nigra*, including the oil obtained from black mulberry seed. The interest in different parts of the *M. nigra* plant is not only for their biological value but also for their economic impact, as antioxidants may be extracted from food by-products or underexploited plant parts. Thus, enhancing the value of black mulberry waste products, such as seeds, for the production of antioxidant-rich ingredients with various health benefits is a sustainable strategy that can support the circular bio-economy and help achieve the zero-waste goal in the food industry [24,25]. Therefore, the present study aimed to provide a comprehensive analysis of the antioxidant and hepatoprotective properties of all aerial parts of the black mulberry plant, including the oil obtained from the seeds.

## 2. Materials and Methods

### 2.1. Chemicals

Acetonitrile, water (HPLC), methylene chloride, ethyl acetate, acetone, ethyl ether, albumin standard, 1-chloro-2,4-dinitrobenzene, nicotinamide adenine dinucleotide phosphate (NADPH) and paraffin wax was obtained from Merck (Germany). Streptozotocin (STZ) was obtained from Sigma Chemicals Co (St. Louis, MO, USA). Referent HPLC standards (chlorogenic acid, vanillic acid, *p*-coumaric acid, rutin, hyperoside, isoquercetin, ellag-ic acid, kaempferol-3-*O*-glucoside, morin, resveratrol, quercetin, luteolin, kaempferol, gallic acid, cyanidin chloride, cynanidin-3-*O*-glucoside, cyanidin-3-*O*-rutinoside, pro-cyanidin B1, protocatechuic acid, protocatechuic acid ethyl ester, *p*-hydroxybenzoic acid, epicatechin and phloridzin) (HPLC grade, ≥99% purity) were obtained from Extrasynthese (Lyon, France). The other chemicals were purchased from Sigma-Aldrich, Schnelldorf, Germany.

### 2.2. Plant Material and Extraction Procedure

Specimens of black mulberry (*Morus nigra* L.) were collected in the Bijeljina region, Bosnia and Herzegovina, in 2015. Voucher specimens (No140716b) were verified and deposited at the Herbarium of the Department of Botany, Faculty of Pharmacy of the University of Belgrade.

The leaves and tree bark were air-dried at room temperature to constant mass. Afterward, they were stored in paper bags, in dry and dark conditions at room temperature. Fresh mature black mulberry fruit was frozen at −21 °C just after harvesting.

Extraction procedures are presented in Table 1.

### 2.3. Chemical Characterization of Black Mulberry Extracts

#### 2.3.1. Determination of Total Phenolic, Tannin, Flavonoid, Anthocyanin and Procyanidin Content in Black Mulberry Extracts

Determination of the total phenolic, tannin and flavonoid content was performed using a UV-VIS Spectrophotometer HP 8453 (Agilent Technologies, Santa Clara, CA, USA).

The total phenolic (TP) content was measured according to the Folin-Ciocalteu method. Absorbance of the mixture was measured at λ_max_ 725 nm. Calibration was performed using gallic acid (0–100 mg/L), with a standard curve linear regression of r^2^ > 0.99. The TP content was expressed as mg of gallic acid equivalents per g of plant extract dry weight (mg GAE/g DW). All measurements were performed in triplicate.

The total tannin (TT) content was determined according to the method presented in the European Pharmacopoeia 9.0 [26]. The absorbance was measured at λ_max_ 760 nm. The percentage of tannins (mean of measurements performed in triplicate) was expressed as pyrogallol (%, *w*/*w*).

The total flavonoid (TF) content was evaluated using the method provided in the European Pharmacopoeia 9.0 [26]. The absorbance was measured at λ_max_ 425 nm. The TF content was expressed as a hyperoside percentage and all measurements were performed in triplicate.

The anthocyanin content of the black mulberry juice was determined after hydrolysis with methanol/HCl under reflux. The absorbance was measured at λ_max_ 528 nm. The percentage of the anthocyanins (mean of measurements performed in triplicate) was expressed as cyanidin-3-glucoside chloride [26].

The procyanidin content of the fruit extract was determined after hydrolysis with methanol/HCl under reflux. The absorbance was measured at λ_max_ 545 nm. The percentage of the procyanidins (mean of measurements performed in triplicate) was expressed as cyanidin chloride [26].

#### 2.3.2. HPLC Analysis of Black Mulberry Extracts

Identification and quantification of the phenolic compounds in the leaf and tree bark extracts was carried out using the 1200 HPLC system (Agilent Technologies, Santa Clara, CA, USA) equipped with a Lichrospher 100 RP 18e, 250 × 4 mm, 5 µm particle size column according to the method of Tadic et al. [28]. The concentrations of the leaf and tree bark ethanol extracts were 27.1 and 33.4 mg/mL, respectively. Prior to injection, the samples were filtered through a PTFE membrane filter. As the standard used in the investigation, the concentration were 0.15 mg/mL for isoquercetin, 0.26 mg/mL for hyperoside, 0.28 mg/mL for kaempferol-3-*O*-glucoside, 0.30 mg/mL for vanillic and kaempferol, 0.34 mg/mL for protocatechuic and gallic acids, 0.36 mg/mL for quercetin, 0.38 mg/mL for resveratrol, 0.40 mg/mL for rutin and epicatechin, 0.56 mg/mL for chlorogenic acid, 0.74 mg/mL for *p*-coumaric acid, 0.25 mg/mL for ellagic acid, 0.17 mg/mL for luteolin and morin, 0.1 mg/mL for procyanidin B1, 0.52 mg/mL for protocatechuic acid ethyl ester and 0.11 mg/mL for *p*-hydroxybenzoic acid and phloridzin. The volume of the standard solutions being injected, as well as for the tested sample extracts, was 4 µL.

The juice and fruit extracts were analyzed for the content of anthocyanins and anthocyanidins according to the method by Ivanovic et al. [29]. The standard solutions for the determination of anthocyanins and anthocyanidins were prepared at a final concentration of 0.2, 0.3 and 0.4 mg/mL (cyanidin chloride, cyanidin-3-*O*-glucoside, cyanidin-3-*O*-rutinoside, respectively) in methanol/HCl. The concentrations of the juice and fruit ethanol extracts were 51.0 and 49.9 mg/mL, respectively. HPLC separation of anthocyanins was achieved using a LiChrospher 100 RP 18e (5 μm), 250 × 4 mm i.d. column with a flow rate of 0.8 mL/min and mobile phase, A [500 mL of H_2_O plus 9.8 mL of 85% H_3_PO_4_(*w*/*w*)], B (ACN), elution, a combination of gradient mode 89–75% A, 0–35 min; 75–60% A, 35–55 min; 60–35% A, 55–60 min; 35–0% A, 60–70 min. Detection was performed using a diode array detector (DAD) and chromatographs were recorded at 520 nm. The standard solutions for the determination of anthocyanins and anthocyanidins were prepared at a final concentration of 0.4, 0.3 and 0.3 mg/mL (cyanidin chloride and cyanidin-3-*O*-glucoside and cyanidin-3-*O*-rutinoside, respectively) in methanol/HCl. The concentrations of the investigated extracts were 11.32 and 22.55 mg/mL for fresh fruit juice and Soxhlet extract of fresh fruit, respectively, in methanol/HCl. The volume of the standard solutions being injected, as well as for the tested sample extracts, was 4 μL. Prior to HPLC analysis, the samples were filtered through a 0.2 μm PTFE filter (Fisher, Pittsburgh, PA, USA).

Identification was based on the retention times and overlay curves. Once spectra matching was achieved, the results were confirmed by spiking with the respective standards to achieve a complete identification by means of the so-called peak purity test. The peaks not fulfilling these requirements were not quantified. Quantification was performed by an external standard method (taking into account the purity of the used standards) and the results were expresses as the mean value ± SD of three measurements.

#### 2.3.3. GC Analysis of Black Mulberry Seeds Oil

Before chemical analysis of the black mulberry seed oil, fatty acids were converted to methyl esters, according to Tadic et al. [28]. The chemical composition of the seed oil was analyzed using the GC and GC/MS technique according to the method by Tadic et al. [28]. A Shimadzu GCMSQP2010 ultra mass spectrometer fitted with a flame ionic detector and coupled with a GC2010 gas chromatograph was used for GC/MS analyses. Separation was performed using an InertCap5 capillary column (60.0 m × 0.25 mm × 0.25 µm). Helium (He), at a split ratio of 1:5 and a linear velocity of 35.2 cm/s, was used as a carrier gas. The temperature of the ion source was 200 °C, the injector temperature was 250 °C and the detector temperature was 300 °C, while the column temperature was linearly programmed from 40 to 260 °C (at a rate of 4 °C/min), from 260 to 310 °C (at a rate of 10 °C/min) and, after reaching 310 °C, kept isothermally for 10 min. Derivatized samples were dissolved in the methylene chloride and injected in an amount of 1 µL.

### 2.4. In Vitro Antioxidant Activity of Black Mulberry Extracts

#### 2.4.1. DPPH Radical Scavenging Activity

DPPH assay was performed according to the method by Tadic et al. [28]. The test solutions were prepared by diluting extracts in ethanol (80 µL of 10 mg/mL extract was diluted in five different concentrations 0.625, 1.25, 2.5, 5.0 and 10.0 mg/mL) or by diluting the black mulberry seed oil in ethanol (to result in the same concentrations). The test solutions (80 µL) were mixed with 2.92 mL ethanol and fresh DPPH ethanolic solution (100 µmol/L). After shaking well, the mixture was incubated for 30 min in the dark at room temperature. The absorbance was recorded at 517 nm, against ethanol as a blank. L-ascorbic acid was used as a positive control.

#### 2.4.2. Ferric-Reducing Antioxidant Power (FRAP)

To determine the antioxidant power of black mulberry extracts, the FRAP assay method was employed according to the methodology presented in the literature [28]. Briefly, 100 µL of extract/oil solutions, as given in the DPPH assay method (diluted extract—1.0 mg/mL and diluted seed oil—1.0 mg/mL) and 3.0 mL of fresh FRAP reagent (25 mL of 300 mM acetate buffer pH 3.6 + 2.5 mL of 10 mM TPTZ solution in 40 mM HCl + 2.5 mL of 20 mM FeCl_3_ × 6H_2_O) were mixed. After 30 min incubation at 37 °C, the absorbance was recorded at 593 nm against a blank. The FRAP value was calculated using a calibration curve of FeSO_4_ × 7H_2_O standard solutions, in the concentration range 100–1000 mmol/L and expressed as mmol Fe^2+^/g extracts. L-ascorbic acid was used as a positive control. Spectrophotometric readings in both assays (DPPH and FRAP) were conducted using an Evolution 60 UV/Vis Spectrophotometer (Thermo Fisher Scientific, Waltham, MA, USA).

### 2.5. Experimental Design and Animal Treatment

Male mice (*Mus musculus*, NMRI Haan strain), with a body weight of 20–30 g, used in the study were provided by the Military Technical Institute, Belgrade. Study approval was obtained from the Ethics Committee on the protection of the welfare of laboratory animals of the University of Novi Sad (Novi Sad, Serbia; No. 01-90/17-1) and the Ministry of Agriculture and Environmental Protection (Belgrade, Serbia; No. 323-07-09649/2015-05). Animal care and all of the experimental procedures were carried out in accordance with the Guide for the Care and Use of Laboratory Animals [30]. During the study, all animals were housed in EHRET airflow Enclosure Volume Power Cabinet Type Uni protect (Ehret, Emmendingen, Germany) with a temperature of 20–25 °C, humidity 55% ± 1.5% and a 12 h light/dark cycle. The animals were provided with free access to food (Veterinary Institute Subotica) and water.

To determine the in vivo antioxidant effects of the described black mulberry extracts, oxidative stress was induced in overnight-fasted mice by a single, intraperitoneal injection of freshly prepared citrate buffer STZ solution (240 mg/kg). Streptozotocin toxicity was confirmed by measuring fasting glucose levels 48 h after STZ administration and animals with FG > 15 mmol/L were included in the experiments.

The animals were divided in seven groups, each containing six animals and treated as described:

1. ConSTZ—saline 10 mL/kg per os for 28 days;

2. ConSTZOil—olive oil 10 mL/kg per os for 28 days;

3. ML-STZ—black mulberry leaf ethanol extract 500 mg/kg per os for 28 days;

4. MB-STZ—black mulberry bark ethanol extract 500 mg/kg per os for 28 days;

5. MF-STZ—black mulberry fruit ethanol extract 500 mg/kg per os for 28 days;

6. MJ-STZ—black mulberry juice 500 mg/kg per os for 28 days;

7. MO-STZ—black mulberry seed oil 500 mg/kg per os for 28 days;

The extracts and juice were dissolved in saline, while the seed oil was dissolved in olive oil adjusted to a final concentration of 500 mg/10 mL.

On the last day of the experiment, the body weight of each animal was measured and the animals were sacrificed by cervical dislocation. Afterward, a necropsy was performed and the liver tissue was collected for further analysis. The weight of the whole liver was measured immediately after removal. Based on the measurements, for each animal, the hepatosomatic index (his) was calculated:(1)$$HSI\;\left(\%\right)=100*\frac{\;liver\;weight\;\left(g\right)}{body\;weight\;\left(g\right)}$$

Samples of liver were taken for histopathological examination and placed in buffered formaldehyde. Samples of liver tissue for determination of the levels of markers of oxidative stress and serum samples were stored at −20 °C until the analysis.

### 2.6. Determination of Markers of Levels of Oxidative Stress

Liver homogenates were prepared by addition of a TRIS buffer (pH X) in 1:4 *w*:*v* and homogenization at 3 °C an electric homogenizer type B, Braun, Potter S (Melsungen, Germany). Samples were centrifuged at 15 min to separate the supernatant and de-nucleated fraction. The protein levels were determined in whole tissue homogenates in supernatant using the biuret test. In whole-tissue homogenates, the levels of malondialdehyde, an indicator of the intensity of lipid peroxidation, were measured as described previously [3]. In the supernatant, activities of the following antioxidant enzymes were determined: catalase (CAT) [31], superoxide-dismutase (SOD) [32], glutathione S-transferase (GST) [33], glutathione reductase (GR) [34] and glutathione peroxidase (GPx) [35].

### 2.7. Histopathology Analysis

After euthanasia, the liver was harvested from each of the experimental animals. Tissue samples (5 × 5 mm) were taken from the right medial liver lobe and fixed in a 10% neutral buffered formalin solution at 4 °C for 24 h. Following fixation, the tissue samples were dehydrated in isopropyl alcohol and embedded in Histowax paraffin blocks (Duiven, The Netherlands). The samples were cut to 5-µm-thick tissue sections on a Leica rotary microtome (Wetzlar, Germany). Hematoxylin and eosin (H&E) and PAS histochemical methods were used for staining the sections. Immunohistochemical staining was performed using the following primary antibodies: anticytochrome P450 2E1 protein (rabbit policlonal, 1:100 dilution, Cusabio, College Park, MD, USA) and anti-Iba1 antibody (rabbit monoclonal, 1:8000 dilution, Abcam, Boston, MA, USA) detected with the UltraVision LP Detection System using HRP Polymer & DAB Chromogen (Thermo Fisher Scientific, Waltham, MA, USA). Before incubation of both antibodies (30 min at room temperature), antigen retrieval was performed using a citrate buffer (pH 6.0) in a microwave oven at 850 W for 20 min. Mayer’s haematoxylin was used as a counterstain for immunohistochemistry prior to mounting and coverslipping (Bio-Optica, Milan, Italy) the slides. The slides were viewed using a Leica DMLB microscope (Wetzlar, Germany) and photographed with a Leica MC 190 HD camera (Wetzlar, Germany) using Leica Live Image Builder software (Wetzlar, Germany). For each PAS and immunohistochemically stained liver histology slide, ten randomly selected microscopic fields of view were photographed at 200× magnification. Afterward, using Fiji morphometry software and a Color Deconvolution plugin, PAS-positive glycogen granules and DAB-positive labeled cells for the anticytochrome P450 marker were isolated from the microphotographs and transferred to a black and white binary format using the threshold function. Within the analyze particles function for black and white photos, based on the contrast, the surface fraction (area fraction) was analyzed, i.e., the percentage occupied by PAS-positive granules and DAB-positive cells in relation to the entire surface of the photograph.

### 2.8. Statistical Analysis

Statistical processing of the data obtained from the test results was done using the statistical software program SPSS (version 21; IBM, Armonk, NY, USA). The arithmetic mean ($\bar{x}$) was used as a measure of the central tendency of a group, while the standard deviation (±SD) expressed the measure of variation among the data. Regression analysis was used to process in vitro results and the strength of correlation was determined by Pearson’s linear correlation coefficient. The intergroup variation was measured by a one-way analysis of variance (ANOVA) followed by Tukey’s post hoc or Kruskal-Wallis with Mann-Whitney post hoc. The results were considered statistically significant when *p*  <  0.05.

## 3. Results

### 3.1. Chemical Analysis

#### 3.1.1. Chemical Analysis of Black Mulberry Extracts (Leaves, Bark, Juice and Fruit)

In the initial screening of the chemical profile of the examined extracts (leaf, bark, juice and fruit), the total phenolic content (TP) was highest in MBEE 68.3 ± 0.7 mgGAE/g (307.1 ± 1.2 mgGAE/g by HPLC method) and MLEE at 23.4 ± 0.5 mgGAE/g (136.4 ± 0.6 mgGAE/g by HPLC method). The total tannin (TT) content was also highest in MBEE at 1.8%, while the total flavonoid content was highest in MLEE at 1.7% (Table 2). The highest content of anthocyanins was identified in MJ 0.84%, while procyanidin content was found to be 1.88% in MFEE.

More than 20 phenolic compounds were identified applying HPLC analysis (Figure 1, Figure 2, Figure 3, Figure 4, Figure 5 and Figure 6). Out of the phenolic acids, predominant were derivatives of hydroxycinnamic acid—chlorogenic acid (142.1 mg/g in MBEE, 40.7 mg/g in MLEE), derivatives of hydroxybenzoic acid—gallic acid (66.2 mg/g in MBEE) and derivatives of dihydroxybenzoic acid—vanillic acid (13.9 mg/g in MBEE). In terms of the flavonoid content, examined black mulberry extracts were most abundant in flavones and flavonols, predominately being isoquercetin (34.8 mg/g in MLEE) and hyperoside (25.9 mg/g in MLEE). Flavanols were identified in MJ (epicatechin, 3.04 mg/g) and MFEE (procyanidin B1, 13.81 mg/g). MJ was also found to be rich in anthocyanins with the highest content of cyanidin chloride, 7.96 mg/g and cynanidin-3-*O*-glucoside, 7.04 mg/g. Furthermore, a bivariate correlation test showed a significant, strong positive correlation between the total phenolics content determined by Folin-Ciocalteu and the HPLC method (Pearson’s r = 0.776; *p* < 0.01).

#### 3.1.2. Chemical Analysis of Black Mulberry Seed Oil

Black mulberry seed oil extracted with supercritical CO_2_ extraction and analyzed with GC/MS was found to be rich in PUFAs. Predominant was the content of linoleic acid and oleic acid esters—methyl linoleate, ω-6 and methyl oleate, ω-9 equaling to 88.5% (Table 3).

### 3.2. Antioxidant Activity of Black Mulberry Extracts (Leaves, Bark, Fruit) Juice and Seed Oil—In Vitro

To evaluate in vitro antioxidant activity, DPPH and FRAP assays were used, in which extracts demonstrated antioxidant activity comparable to the positive control. In DPPH assay, MLEE and MBEE demonstrated the highest antioxidant activity with IC_50_ values of 7.46 µg/mL and 8.62 µg/mL, respectively (ascorbic acid 4.69 µg/mL). Similar results were obtained in FRAP assay. Antioxidant activity was highest in MBEE, MLEE and MSO and resulted in approximately 70% of the antioxidant activity of ascorbic acid (Table 4).

### 3.3. Antioxidant Activity of Black Mulberry Extracts (Leaves, Bark, Fruit) Juice and Seed Oil—In Vivo

The antioxidant activity of black mulberry extracts was evaluated in vivo after STZ administration in NMRI Haan mice, followed by 28-day treatment with examined extracts.

#### 3.3.1. Hepatosomatic Index

In all experimental groups, the hepatosomatic index average value was lower compared to the corresponding control group. The lowest values were obtained in groups of animals treated with MBEE (0.052 ± 0.005) and MJ (0.053 ± 0.002) (Table 5).

#### 3.3.2. Activity of Antioxidant Enzymes

The antioxidant activity of the tested extracts was further evaluated by measuring the activity of the antioxidant enzymes that represent the defense mechanism in ROS-induced injury. The extracts with the highest antioxidant effect determined in vivo were MLEE, MBEE and MSO.

The intensity of lipid peroxidation, evaluated through MDA level, was lower in all experimental groups when compared to the respective controls. A significantly lower intensity of lipid peroxidation was found in the groups treated with MLEE and MBEE (ML-STZ 0.043 ± 0.009 *; MB-STZ 0.045 ± 0.007 *, ConSTZ 0.056 ± 0.021), as well as in the group treated with MSO (MO-STZ 0.039 ± 0.016 *; ConOilLSTZ 0.049 ± 0.021) (*p* < 0.05) (Figure 7).

The activity of the enzyme superoxide dismutase (ML-STZ 26.17 ± 2.98; MB-STZ 28.69 ± 3.66, ConSTZ 19.98 ± 2.26) and catalase (ML-STZ 33.46 ± 2.19; MB-STZ 32.86 ± 3.67; ConSTZ 27.00 ± 2.36) was significantly higher in groups of animals treated with leaf and bark extracts compared to the control group treated with saline (* *p* < 0.05). A similar effect was determined in the group of animals treated with MSO, with a significantly higher activity of superoxide dismutase compared to the control group (MO-STZ 27.36 ± 2.72 *; ConOilSTZ 22.14 ± 2.44; * *p* < 0.05), while catalase activity was higher, but the difference was not statistically significant (MO-STZ 32.35 ± 2.49; ConOilSTZ 29.83 ± 2.09) (Figure 7).

The activities of glutathione peroxidase, glutathione reductase and glutathione-S-transferase enzymes were similar or higher in the experimental groups treated with extracts of black mulberry and seed oil compared to the corresponding control groups, but with differences being insignificant (Figure 7).

#### 3.3.3. Histopathology Analysis

The hepatoprotective effect of the tested extracts was verified in histopathology evaluation as well (Figure 8). In histological analysis of the liver tissue in all control and experimental groups, no visible morphological disturbances in the structure were observed with standard H&E staining, (images A, E, I, M, Q, U and Y). Hepatocytes were characterized by a slightly granular cytoplasm and a regular cytoarchitectonic arrangement of rows of hepatocytes (Remak trabeculae) oriented towards the vena centralis. The portal spaces were regularly positioned without visible inflammatory cells.

Using histochemical PAS staining, a higher concentration of PAS-positive material inside hepatocytes (most likely glycogen) in groups such as ML-STZ, MB-STZ, as well as to a lesser degree in MO-STZ (images F, J and Z) was observed compared to the remaining groups (images B, N, R and V).

Immunohistochemical staining for Iba1, used to visualize tissue macrophages (Kupffer cells), did not indicate differences regarding the change in the number and distribution of macrophages between the examined groups (images C, G, K, O, S, W and ZZ).

However, an immunohistochemical analysis of P450 enzyme activity indicated a reduced positivity in the ML-STZ and MB-STZ groups (images H and L) compared to the remaining groups (images D, P, T, X and YY).

The comparison of the mean values of the ranks of the surface fractions of PAS-positive granules and DAB-positive cells between the control and experimental groups was performed using the Kruskal-Wallis test. PAS positivity was significantly higher in the MB-STZ group in comparison to all other groups. Immunohistochemistry of P450 showed significantly lower positivity in the ML-STZ and MB-STZ groups compared to all other experimental groups (Figure 9).

## 4. Discussion

The results of our study showed that examined extracts were rich in phenolic compounds and the obtained results are comparable to the results presented in other studies [36,37,38].

Like other polyphenolic compounds, phenolic acids exhibit antioxidant effects and the most abundant in plants are chlorogenic, gallic, vanillic and *p*-coumaric acids [39,40]. The richest content of phenolic acids was identified in MBEE. Chlorogenic acid was the most prominent one (142.1 ± 2.1 mg/g) followed by gallic acid (66.2 ± 0.9 mg/g). Chlorogenic acid is regarded as, if not the major, one of the major phenolic constituents in *Morus nigra* leaves [40], which was also confirmed in our study with MLEE being most abundant in chlorogenic acid (40.7 ± 0.7 mg/g).

Among flavones and flavonols, the most abundant in analyzed extracts were quercetin, kaempferol, isoquercetin, hyperoside and rutin. Most of these substances were identified in the highest concentration in MLEE (isoquercetin 34.8 mg/g; hyperoside 25.9 mg/g), which is in line with other studies suggesting that flavones and flavonols are among the most important phenolic compounds in black mulberry leaves [41]. Although several studies identified rutin as the most relevant flavonoid in black mulberry fruits [12,18,42], the presence of rutin was detected only in MLEE with 2.7 mg/g of extract.

The pigments of anthocyanins are primarily isolated from black mulberry fruits and leaves [11,41]. These compounds demonstrate strong antioxidant effects and the potential to be regarded as functional foods consumed for the prevention of various diseases [12,43,44]. In our study, the most abundant anthocyanidin in MFEE was cyanidin chloride (7.96 ± 0.61 mg/g), followed by cyanidin-3-*O*-glucoside (7.04 ± 0.54 mg/g) and cyanidin-3-*O*-rutinoside (2.10 ± 0.11 mg/g). These results are in agreement with other studies referring to *M. nigra* as an anthocyanins-rich fruit species [45].

Regarding the composition of black mulberry seed oil, there is a limited number of papers suggesting that the composition of black mulberry seed oil as well as its potential therapeutic effects are insufficiently studied. According to available literature, the composition of the seed oil revealed a high content of fatty acids, particularly linoleic, followed by oleic, palmitic and stearic acids [46,47].

The levels of most fatty acids detected in our study are consistent with the results of previously published studies using whole black mulberry fruit [46]. In the tested MSO used in our study, the PUFAs (linoleic and oleic), precisely methyl esters of linoleic and oleic acid, were the most abundant, with a total share of 88.5%, followed by palmitic 8.6%, stearic 2.5% and arachidonic 0.1%. Hexane Soxhlet extraction of black mulberry seed oil reported by Getzgel et al. and Yilmaz et al. resulted in a lower share of linoleic and oleic fatty acids (76.61% [46] and 83.37% [47]) compared to our results obtained when the supercritical CO_2_ extraction was employed, suggesting that supercritical CO_2_ extraction affected the content of polyunsaturated fatty acids in oil. The other matter that needs to be mentioned is the environmental aspect of extraction, making supercritical CO_2_ extraction the method of choice. Unlike supercritical CO_2_ extraction, which is observed as environmentally friendly, Soxhlet extraction generates contaminated, hazardous solvents and emits toxic fumes [48].

Furthermore, the antioxidant activity of the examined extracts and oil was tested in vitro using DPPH and FRAP assay. As expected, based on the high TP content, in both assays, MBEE and MLEE stood out among the other extracts regarding their antioxidant activity. The total content of phenolic compounds in examined extracts, determined both by Folin-Ciocalteu and HPLC method, showed a linear correlation with the antioxidant capacity (Pearson’s r = 0.776; *p* < 0.01), confirming that antioxidant activity was dependent on the TP content. Similar results regarding leaf and bark antioxidant activity were published by other study groups using DPPH [20,49,50] or FRAP assay [42,51,52]. The antioxidant activity of MBEE was likely based on the rich content of phenolic acids (chlorogenic acid, gallic acid), which are known to possess antioxidant activity [53,54,55], while flavonoids (hyperoside, isoquercetin) also being recognized for their antioxidant effect [56,57,58], contributed to the antioxidant activity of MLEE.

MSO also showed a pronounced antioxidant capacity according to the performed tests, although lower in DPPH assay, but almost identical to the values determined for MBEE and MLEE in FRAP assay. Most probably, the antioxidant effect of MSO was based on the rich content of polyunsaturated fatty acids, which was higher compared to other studies [46,47]. It is known that unsaturated fatty acids possess antioxidant activity and are regarded as beneficial for health [59,60]. Although ω-3 tends to be superior to ω-6, ω-6, unsaturated fatty acids such as linoleic and oleic exhibit numerous positive pharmacological effects mediated by antioxidant activity, e.g., cardioprotective, hypolipidemic, antiatherosclerotic effects, etc. [61,62].

The antioxidant capacity of the examined extracts was also confirmed in vivo, in the model of STZ-induced oxidative stress in mice. Namely streptozotocin, a derivative of D-glucosamine, is an antibiotic capable of inducing oxidative stress, which not only damages pancreatic tissue resulting in diabetes, but also other GLUT 2-expressing organs such as the liver, kidney and brain [63,64,65,66,67]. Free radicals that accompany hyperglycemia lead to damage of various tissues, especially blood vessels, and they are regarded as significant contributors to atherogenesis. A disturbed balance between pro-oxidative agents and the body’s antioxidant defense (glutathione-peroxidase (GPd), superoxide-dismutase (SOD) and catalase (CAT)) is considered to be the main factor in the pathogenesis of micro- and macrovascular complications that occur in patients with diabetes [68,69].

In animals treated with MLEE, MBEE and MSO, the level of malonylaldehyde, a marker for the intensity of lipid peroxidation, was decreased, suggesting that the administered extracts increased the capacity of antioxidant protection. On the other hand, SOD and CAT activity was significantly higher in groups of animals treated with MLEE and MBEE compared to the control group. A similar pattern was identified in the group treated with MSO. The observed simultaneous increase in the activity of both these enzymes is particularly significant. Superoxide dismutase and catalase are enzymes of the first line of cell protection against oxidative stress [70,71]. Under physiological conditions, the superoxide radical, a product of reactions in cellular metabolism, is neutralized under the action of superoxide dismutase. In this process, hydrogen peroxide is produced, which is broken down by the CAT enzyme (glutathione-peroxidase in the mitochondria). In the case of pronounced oxidative stress, with an increased amount of superoxide radicals, the increase in SOD activity enables the neutralization of superoxide to oxygen and less toxic hydrogen peroxide. The resulting hydrogen peroxide serves as a precursor for the formation of other, much more reactive free radicals, therefore its timely decomposition requiring adequate activity of CAT is necessary. Reduction of CAT activity is recognized as one of the factors that contribute to the occurrence of damage mediated by oxidative stress in diabetes and other degenerative diseases [72]. The ability of the tested extracts to increase the activity of both mentioned enzymes indicates a significant potential for adequate antioxidant protection in conditions with increased oxidative stress, which is in line with results of other studies [67,73]. We must also point out that there were no significant differences related to the activity of other enzymes, suggesting that the intensity of streptozotocin-induced oxidative stress was moderate, affecting only the enzymes of the first line of cell protection against oxidative stress.

The protective antioxidant effect was also tested via the hepatosomatic index and histological analysis of liver tissue. The hepatosomatic index, defined as the ratio between liver weight and body weight, can be an indicator of liver damage and inflammation [74]. However, it was not completely proven in our study. To be specific, in groups treated with MBEE, MJ and MSO, the hepatosomatic index was lower (suggesting less liver damage and inflammation) compared to the referent control groups (ConSTZ and MB-STZ, MJ-STZ; ConOilSTZ and MO-STZ), but the difference was insignificant. On the other hand, in histopathological analysis, treatment with black mulberry extracts resulted in moderate signs of hepatoprotective effect in STZ-induced liver injury. Given that the tissue changes that occurred after STZ administration were most likely reversible, the preservation of liver tissue in all examined groups observed in HE staining was expected. In addition, immunohistochemical staining for Iba1 did not indicate differences regarding the change in the number and distribution of macrophages. However, administration of black mulberry extracts in the ML-STZ and MB-STZ as well as in MO-STZ groups resulted in a higher restitution of glycogen energy reserves detected in PAS staining, with statistical significance being detected only in the MB-STZ group (Figure 9). Additionally, the significantly weaker activity of the P450 enzyme was detected in the ML-STZ and MB-STZ groups (Figure 9), which could be an indirect indicator of a better recovery of liver tissue [3].

Other models of liver damage provided similar results. The hepatoprotective effect of black mulberry leaf extract was confirmed in a model of paracetamol-induced liver damage, where the administration of high doses of paracetamol leads to oxidative stress and consequent structural and functional abnormalities of the hepatobiliary tract [75]. The protective effect of the leaf extract was also confirmed in the study by Agha et al. [76], in which black mulberry fruit juice reduced liver damage induced by carbon tetrachloride.

Overall, it is believed that the hepatoprotective effect is based on the neutralization of free radicals and the antioxidant activity of black mulberry components such as phenolic acids (chlorogenic acid, gallic acid) and flavonoids (quercetin, rutin and isoquercetin) [53,54,55,77,78,79]. Besides MLEE and MBEE, MSO was also found to possess an antioxidant effect in an in vivo model, likely mediated by fatty acids, which are known to possess strong antioxidant, anti-inflammatory and analgesic effects [45,80].

## 5. Conclusions

The findings of this research suggest that black mulberry represents a rich source of bioactive components that help counteract compounds with pro-oxidative action. Antioxidant activity was confirmed by both in vitro and in vivo methods. According to the available publications, this is the first study in which the effect of all aerial parts of the black mulberry plant (leaf, bark, fruit, juice, seeds) were examined in one study. Due to the higher content of phenolic compounds, leaf and bark ethanol extracts were found to possess a stronger antioxidant and hepatoprotective effect.

It should be emphasized that there is no available data on the effects and use of black mulberry seeds, more precisely on the oil obtained from the seeds. This study revealed that black mulberry seed oil was characterized by significant antioxidant activity, likely mediated by a rich fatty acid content. The added value of the present study is related to the method of oil extraction from black mulberry seeds, which is based on the use of “green technology”—extraction with supercritical CO_2_, where no organic solvents were used and there were no waste products [81,82,83].

On the other hand, this study has some limitations. Although the presumed main mechanism of action of tested extracts is direct ROS scavenging activity, the limitation of the presented study would be a lack of the results on the molecular mechanism of action. In addition, the in vivo model of streptozotocin-induced oxidative stress resulted in inconclusive effects in some of the tested parameters. Future studies would be of benefit to test the antioxidant properties (including the mechanism of action) of the most potent black mulberry extracts, primarily of seed oil, in different liver injury models (therapeutic purposes), as well as to test its prophylactic use. Due to the dynamic nature of antioxidant enzyme activity, it would also be beneficial to test the antioxidant activity of black mulberry extracts in vivo in different time points after the onset of liver injury to evaluate acute and chronic use effects.

## Figures and Tables

**Figure 1 antioxidants-12-00382-f001:**
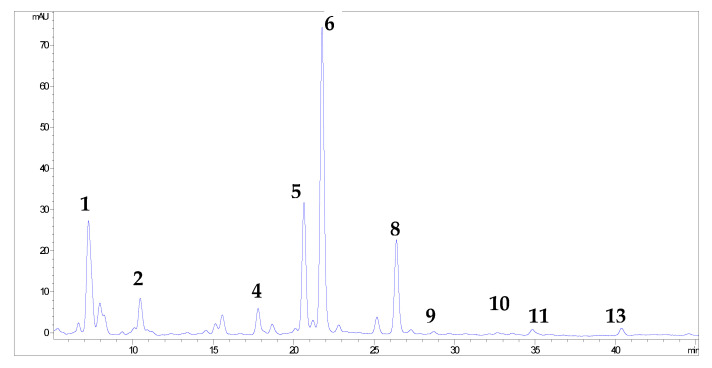
HPLC chromatograms of investigated black mulberry leaf ethanol extract and phenolic compounds identified (the numbers refer to corresponding peaks, presented in Table 1): chlorogenic acid (1), vanillic acid (2), rutin (4), hyperoside (5), isoquercetin (6), kaempferol-3-O-glucoside (8), phloridzin (9), morin (10), resveratrol (11) and luteolin (13).

**Figure 2 antioxidants-12-00382-f002:**
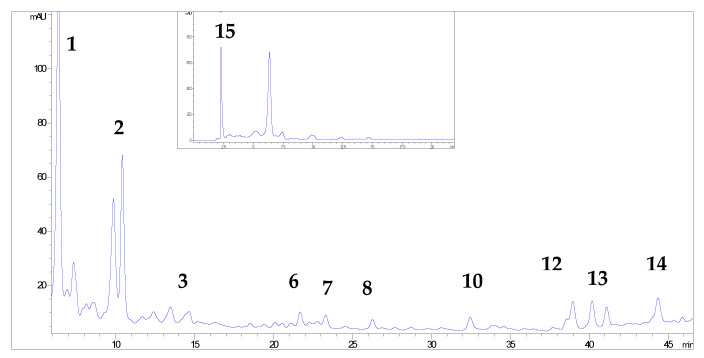
HPLC chromatograms of investigated black mulberry tree bark ethanol extract and phenolic compounds identified (the numbers refer to corresponding peaks, presented in Table 1): chlorogenic acid (1), vanillic acid (2), p-coumaric acid (3), isoquercetin (6), ellagic acid (7), kaempferol-3-O-glucoside (8), morin (10), quercetin (12), luteolin, (13) kaempferol (14) and gallic acid (15).

**Figure 3 antioxidants-12-00382-f003:**
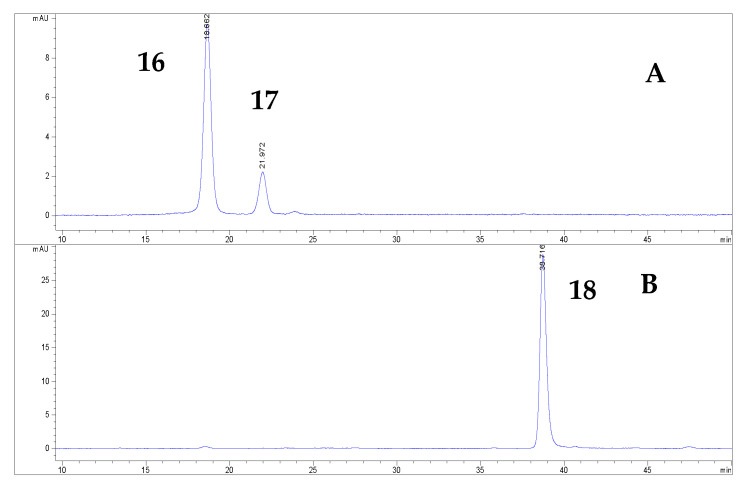
HPLC chromatograms of investigated black mulberry juice and anthocyanin compounds identified at 520 nm (**A**) before hydrolysis (the numbers refer to corresponding peaks, presented in Table 1): cyanidin-3-O-glucoside (16), cyanidin-3-O-rutinoside (17); (**B**) after hydrolysis: cyanidin chloride (18).

**Figure 4 antioxidants-12-00382-f004:**
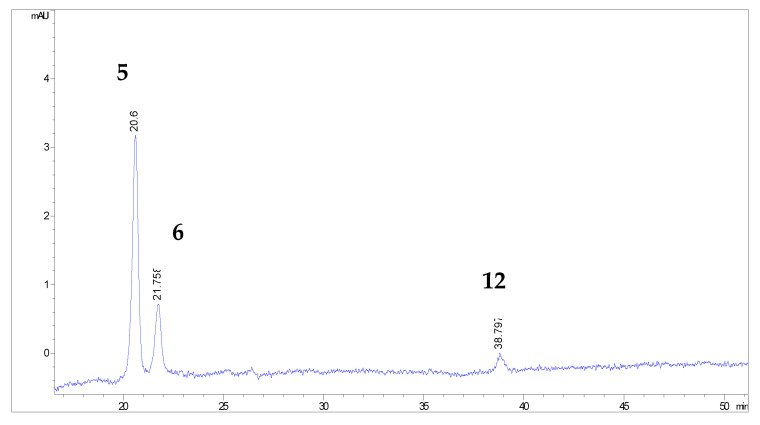
HPLC chromatograms of investigated black mulberry juice and flavonoid compounds identified (the numbers refer to corresponding peaks, presented in Table 1): hyperoside (5), isoquercetin (6) and quercetin (12).

**Figure 5 antioxidants-12-00382-f005:**
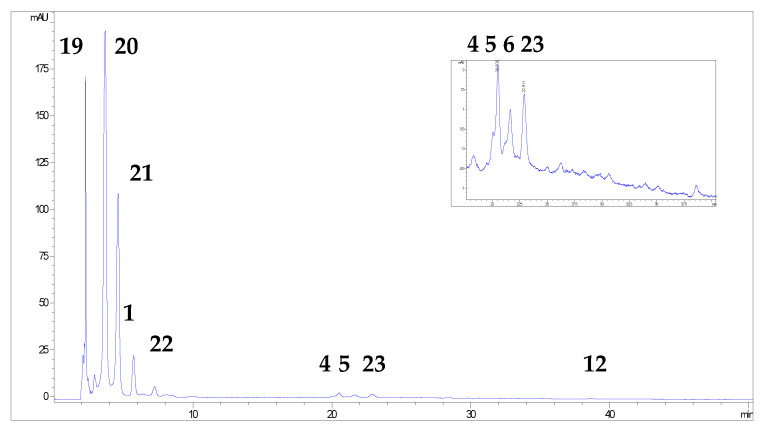
HPLC chromatograms of investigated black mulberry fruit ethanol extract and phenolic compounds identified (the numbers refer to corresponding peaks, presented in Table 1): procyanidin B1 (19), protocatechuic acid (20), p-hydroxybenzoic acid (21), chlorogenic acid (1), epicatechin (22), rutin (4), hyperoside (5), isoquercetin (6), protocatechuic acid ethyl ester (23) and quercetin (12).

**Figure 6 antioxidants-12-00382-f006:**
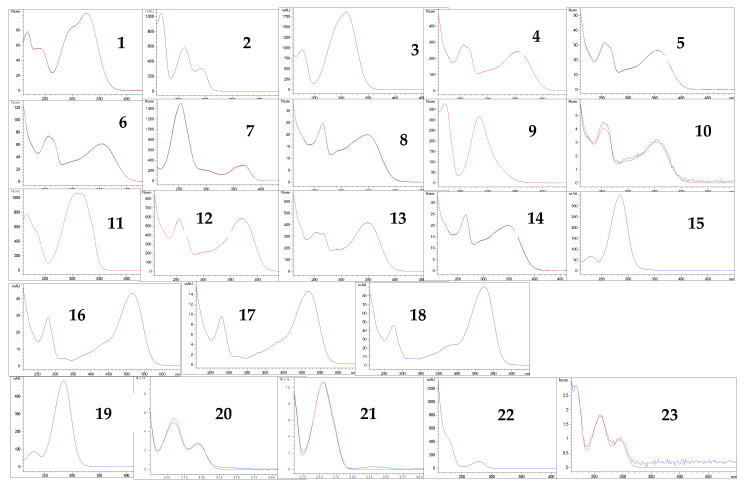
UV spectra of identified compounds present in the investigated samples (the numbers refer to corresponding peaks, presented in Table 2).

**Figure 7 antioxidants-12-00382-f007:**
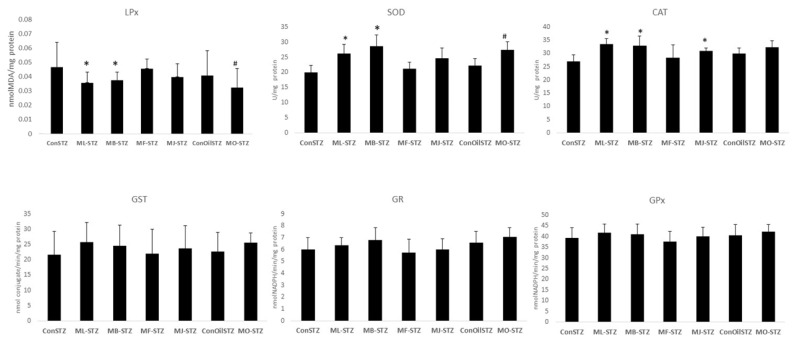
Effect of mulberry extracts on the levels of markers of oxidative stress and antioxidant enzymes activity in liver tissue homogenates in mice with streptozotocin-induced oxidative stress (ConSTZ—saline 10 mL/kg; ConSTZOil—olive oil 10 mL/kg; ML-STZ—black mulberry leaf ethanol extract; MB-STZ—black mulberry bark ethanol extract; MF-STZ—black mulberry fruit ethanol extract; MJ-STZ—black mulberry juice; MO-STZ—black mulberry seed oil) (LPx—lipid peroxidation; SOD—superoxid dismutase; CAT—catalase, GST—glutathione S-transferase; GR—glutathione reductase; GPx—glutathione peroxidase). * significantly different compared to ConSTZ group; # significantly different compared to ConOilSTZ group (vehicle control).

**Figure 8 antioxidants-12-00382-f008:**
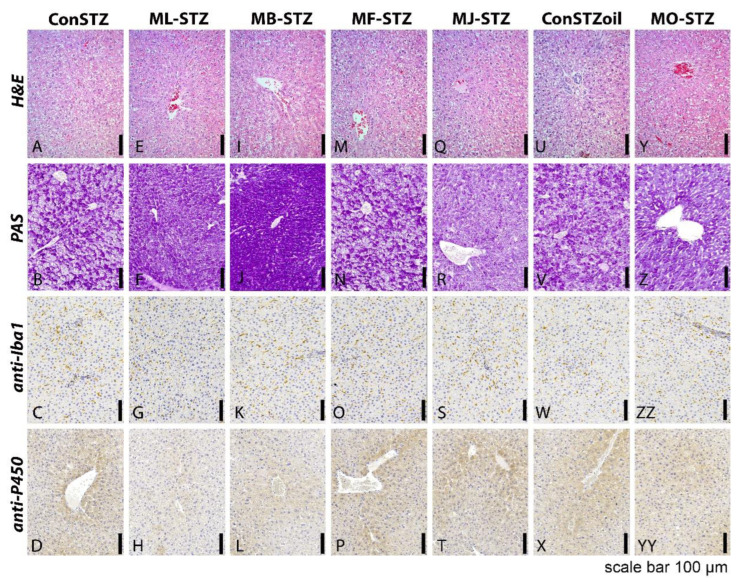
Histopathology analysis of the effects of black mulberry extracts in streptozotocin-induced liver injury (ConSTZ—saline 10 mL/kg; ConSTZOil—olive oil 10 mL/kg; ML-STZ—black mulberry leaf ethanol extract; MB-STZ—black mulberry bark ethanol extract; MF-STZ—black mulberry fruit ethanol extract; MJ-STZ—black mulberry juice; MO-STZ—black mulberry seed oil). H&E (**A**–**Y**), PAS (**B**–**Z**); anti-Iba1 (**C**–**ZZ**); anti-P450 (**D**–**YY**); magnification 200×; scale bar 100 µm.

**Figure 9 antioxidants-12-00382-f009:**
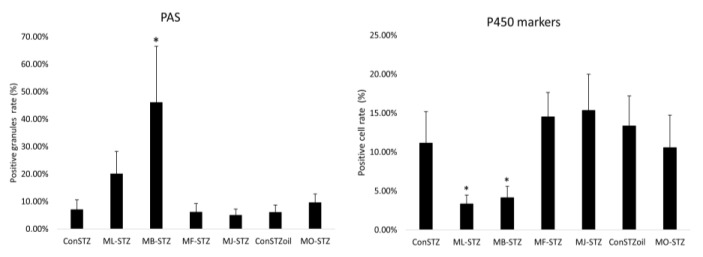
Quantitative (PAS and P450 markers) histopathology analysis of the effects of black mulberry extracts in streptozotocin-induced liver injury. (* *p* < 0.001 vs. ConSTZ).

**Table 1 antioxidants-12-00382-t001:** Black mulberry extraction procedures.

Extract	Extraction Procedure	Extraction Yield	Ref.
Black mulberry leaf extract (MLEE)	Soxhlet ethanol extraction procedure—milled dry black mulberry leaves (Braun Aromatic KS2 mill) and 70% *v*/*v* ethanol (ethanol:distilled water 70:30). The extract was prepared using 70 g of dry leaves and 1000 mL of ethanol. After five-day extraction, the extract was filtered and evaporated to dry on a rotary evaporator at 50 °C (Buchi, Switzerland)	52.13%	[26]
Black mulberry tree bark extract (MBEE)	Soxhlet ethanol extraction procedure—milled dry black mulberry leaves (Braun Aromatic KS2 mill) and 70% *v*/*v* ethanol (ethanol:distilled water 70:30). The extract was obtained by treating 70 g dry tree bark with 1000 mL of 70% ethanol. After seven-day extraction, the extract was filtered and evaporated to dry on a rotary evaporator at 50 °C (Buchi, Switzerland).	23.63%	[26]
Black mulberry fruit extract (MFEE)	Soxhlet ethanol extraction procedure. The extract was prepared with 300 g of defrosted fruit and 450 mL of 70% *v*/*v* ethanol (ethanol:distilled water 70:30). After seven-day extraction, the extract was filtered and evaporated to dry on a rotary evaporator at 50 °C (Buchi, Switzerland).	60.17%	[26]
Black mulberry juice (MJ)	The juice was prepared manually by squeezing the fresh mature black mulberry fruit through a sieve. The fresh juice was frozen at −21 °C until the beginning of the study.	-	
Black mulberry seed oil (MSO)	Black mulberry seeds were obtained from fruit pulp after squeezing out the juice from the fruit. The seeds were air-dried at room temperature to constant mass. The seed oil was extracted using a supercritical CO_2_ extraction procedure in an Autoclave Engineers Screening system. The plant material (60 g) was placed in the extractor vessel and the system was heated to the desired temperature. Afterward, CO_2_ was introduced using a liquid metering pump until the required pressure was obtained. The extract was collected in the separator and its mass was measured at certain time intervals to determine the extraction yield. The oil extraction was performed at 300 bar pressure and 40 °C temperature. The CO_2_ consumption during extraction was 50 kg CO_2_/kg with a constant flow rate of 0.5 kg/h. After extraction, the oil was stored at −20 °C until analyzed.	17.10%	[27]

**Table 2 antioxidants-12-00382-t002:** Content of phenolic compounds identified in black mulberry leaf ethanol extract (MLEE), black mulberry bark ethanol extract (MBEE), black mulberry juice (MJ) and black mulberry fruit ethanol extract (MFEE) (mean of measurements performed in triplicate ± SD).

	Compounds	MLEE	MBEE	MJ	MFEE
	Total Tannins (%)	0.4 ± 0.1	1.8 ± 0.1	0.05 ± 0.01	0.52 ± 0.01
	Anthocyanins (%)	/	/	0.84 ± 0.01	0.11 ± 0.01
	Procyanidin content (%)	/	/	-	1.88 ± 0.02
	Total flavonoids (%)	1.7 ± 0.1	0.5 ± 0.1	/	/
	Content of polyphenol (mgGAE/g dry residue) *	23.4 ± 0.5 ^1^	68.3 ± 0.7 ^1^	14.69 ± 0.09 ^1^	18.68 ± 0.11 ^1^

	**Compound (mg/g extract)**				
1	chlorogenic acid	40.7 ± 0.7	142.1 ± 2.1	/	0.51 ± 0.06
2	vanillic acid	1.3 ± 0.1	13.9 ± 0.2	/	/
3	*p*-coumaric acid	/	2.8 ± 0.1	/	/
4	rutin	2.7 ± 0.1		/	/
5	hyperoside	25.9 ± 0.2		0.1 ± 0.01	0.03 ± 0.00
6	isoquercetin	34.8 ± 0.5	2.6 ± 0.1	0.07 ± 0.01	0.02 ± 0.00
7	ellagic acid		0.8 ± 0.1	/	/
8	kaempferol-3-*O*-glucoside	4.2 ± 0.1	0.5 ± 0.1	/	/
9	phloridzin	t	/	/	/
10	Morin	/	0.9 ± 0.1	/	/
11	resveratrol	1.0 ± 0.1		/	/
12	quercetin		2.4 ± 0.1	0.01 ± 0.00	/
13	luteolin	0.3 ± 0.0	2.2 ± 0.1	/	/
14	kaempferol	/	2.1 ± 0.1	/	/
15	gallic acid	/	66.2 ± 0.9	/	/
16	cyanidin chloride	/	/	7.96 ± 0.61	/
17	cynanidin-3-*O*-glucoside	/	/	7.04 ± 0.54	/
18	cyanidin-3-*O*-rutinoside	/	/	2.10 ± 0.11	/
19	procyanidin B1	/	/	/	13.81 ± 0.12
20	protocatechuic acid	/	/	/	1.33 ± 0.11
21	*p*-hydroxybenzoic acid	/	/	/	0.1 ± 0.00
22	epicatechin	/	/	3.04 ± 0.02	0.1 ± 0.00
23	protocatechuic acid ethyl ester			/	0.03 ± 0.01
	Total phenolics by HPLC	136.4 ± 0.6 ^1^	307.1 ± 1.2 ^1^	35.9 ± 0.3 ^1^	18.44 ± 0.4 ^1^

* determined by Folin-Ciocalteu method; ^1^
*p* < 0.001 vs. all other groups.

**Table 3 antioxidants-12-00382-t003:** Fatty acid composition of black mulberry seed oil (MSO).

Compound	RI—Retention Index	%
Methyl palmitate	1925	8.6
Methyl linoleate, ω-6	2085	87.1
Methyl oleate, ω-9	2101	1.4
Methyl stearate (methyl octadecanoate)	2123	2.5
Eicosanoic acid methyl ester (methyl arachidonate)	2134	0.1
Total		99.7

**Table 4 antioxidants-12-00382-t004:** Percentage of DPPH inhibitory values and IC50 values and FRAP antioxidant capacity of black mulberry leaf ethanol extract, black mulberry bark ethanol extract, black mulberry juice, black mulberry fruit ethanol extract and black mulberry seed oil (mean of measurements performed in triplicate ± SD).

Sample	DPPH IC_50_ (µg/mL)	FRAP mmolFe (II)/g of Extract
Ascorbic acid	4.69 ± 0.4 ^1,2,3,4^	1.389 ± 0.1 ^1,3^
MLEE	7.64 ± 0.68 ^1,2^	0.97 ± 0.21
MBEE	8.62 ± 0.72 ^1,2^	0.989 ± 0.20
MFEE	17.99 ± 0.99	0.784 ± 0.30
MJ	10.49 ± 0.93 ^1,2^	0.499 ± 0.45
MSO	24.19 ± 1.15	0.96 ± 0.21

^1^*p* < 0.01 vs. MFEE; ^2^
*p* < 0.01 vs. MSO; ^3^
*p* < 0.05 vs. MBEE; ^4^
*p* < 0.01 vs. MJ.

**Table 5 antioxidants-12-00382-t005:** Hepatosomatic index in mice with streptozotocin-induced oxidative stress (ConSTZ—saline 10 mL/kg; ConSTZOil—olive oil 10 mL/kg; ML-STZ—black mulberry leaf ethanol extract; MB-STZ—black mulberry bark ethanol extract; MF-STZ—black mulberry fruit ethanol extract; MJ-STZ—black mulberry juice; MO-STZ—black mulberry seed oil).

Group	x ± SD
ConSTZ	0.058 ± 0.01
ML-STZ	0.057 ± 0.02
MB-STZ	0.052 ± 0.005
MF-STZ	0.055 ± 0.006
MJ-STZ	0.053 ± 0.002
ConOilSTZ	0.062 ± 0.006
MO-STZ	0.059 ± 0.013

## Data Availability

Data available on a reasonable request from authors.

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
