# Peer review of "Comprehensive Analysis of Antioxidant and Hepatoprotective Properties of Morus nigra L."

_antioxidants, 2023, doi:10.3390/antiox12020382_

Round 1
Reviewer 1 Report
The study of Saša Vukmirović et al., is a well described study regarding the beneficial effects of extracts of Morus nigra performed from diferent parts of the plant. The novelty of this study is based on the promotion of circular economy and the reduction of plant waste and the strongest part of the manuscript is the comprehensive analysis of antioxidant properties of the extracts of different parts, oil and juice.
However, there are some issues that need to be addressed.
1.In table 1 it seems that statististical analysis is missing. Moreover, authors need to explain whether SEM or standard deviation is included and whether the analysis is made in duplicate or triplicate.
2. The same in table 3. Also, statistical analysis seems to be missing. Some of the values are quite different. What about SEM and/or STDEV?
3. For the calculation of hepatosomatic index, it is important to show that there are no differences between body weightsof the animals. Otherwise it is not clear whether liver is greater in one group than another.
4.Images should be analyzed to measure liver damage. Images of the IHC are dificult to be appreciated. Is there any analysis of the images?
5. Discussion is centered in antioxidant properties of the Morus migra L. however hepatoprotective effects are not clear, in the majority of the enzymes, no effect is observed. However authors do not discuss openly these results. The same applies with the conclusions.
Reviewer 2 Report
The manuscript reports a comprehensive profile of a series of Morus nigra extracts with emphasis on the antioxidant properties. Chemical analysis and antioxidant activity both in vitro and in vivo were determined.
The manuscript presents several flaws. The main problem is related to the fact that that biological effects are only reported for the full extract without identification of the compounds responsible for the observed effects and no mechanism of action is provided either. In addition, I have serious doubts on peak identification (no chromatogram was shown)! A total of 22 were quantified. How? No reference to it! Only 15 reference standards (and not Referent HPLC standards, line 84) were employed. How it is possible?
I cannot see any relevant novelty aspects on the topic since many studies have already shown that black mulberries are rich in bioactive compounds and numerous biological and pharmacological activities have been documented. The only exception is the analysis and evaluation of the oil obtained from black mulberry seed but is does not justify publication on a high-impact factor journal like Antioxidants.
The language is poor and does require a deep scientific revision.
Title: It is not suitable for a scientific journal!
Abstract: Analytical methodologies employed are not mentioned!
Introduction: it is too maigre and confusing.
2.2. Plant material and extraction procedure.
It is not clear how many samples were considered.
It is not clear whether validated protocols have been employed.
Round 2
Reviewer 1 Report
Authors have adressed correctly the comments of the reviewers.
A small note: in table 2 samples analyzed were duplicates or triplicates.
Author Response
Response to Reviewer 1 Comments
Thank you for taking time to review our manuscript during both rounds and for your comments on our study, which helped us to improve the content and overall quality of the manuscript.
Point 1. A small note: in table 2 samples analysed were duplicates or triplicates.
Response 1. Samples analysed and presented in table 2 were triplicates. In order to resolve this issue, we have added following text in the table caption:
“(mean of measurements performed in triplicate ± SD)”
Reviewer 2 Report
The authors have adequately addressed all remarks. Only a little note: at line 477, it should be read: Figure 5.
Author Response
Response to Reviewer 2 Comments
Thank you for taking time to review our manuscript during both rounds and for your detailed and highly insightful comments, which helped us to improve the content and overall quality of the manuscript.
Reviewer 2
Point 1. Only a little note: at line 477, it should be read: Figure 5.
Response 1. At line 477 we inserted Figure 5. instead of Figure x.